# Oxidative Metabolism as a Cause of Lipid Peroxidation in the Execution of Ferroptosis

**DOI:** 10.3390/ijms25147544

**Published:** 2024-07-09

**Authors:** Junichi Fujii, Hirotaka Imai

**Affiliations:** 1Department of Biochemistry and Molecular Biology, Graduate School of Medical Science, Yamagata University, Yamagata 990-9585, Japan; 2Laboratory of Hygienic Chemistry, School of Pharmaceutical Sciences, Kitasato University, Tokyo 108-8641, Japan; 3Medical Research Laboratories, School of Pharmaceutical Sciences, Kitasato University, Tokyo 108-8641, Japan

**Keywords:** glycolysis, tricarboxylic acid cycle, urea cycle, metabolic remodeling

## Abstract

Ferroptosis is a type of nonapoptotic cell death that is characteristically caused by phospholipid peroxidation promoted by radical reactions involving iron. Researchers have identified many of the protein factors that are encoded by genes that promote ferroptosis. Glutathione peroxidase 4 (GPX4) is a key enzyme that protects phospholipids from peroxidation and suppresses ferroptosis in a glutathione-dependent manner. Thus, the dysregulation of genes involved in cysteine and/or glutathione metabolism is closely associated with ferroptosis. From the perspective of cell dynamics, actively proliferating cells are more prone to ferroptosis than quiescent cells, which suggests that radical species generated during oxygen-involved metabolism are responsible for lipid peroxidation. Herein, we discuss the initial events involved in ferroptosis that dominantly occur in the process of energy metabolism, in association with cysteine deficiency. Accordingly, dysregulation of the tricarboxylic acid cycle coupled with the respiratory chain in mitochondria are the main subjects here, and this suggests that mitochondria are the likely source of both radical electrons and free iron. Since not only carbohydrates, but also amino acids, especially glutamate, are major substrates for central metabolism, dealing with nitrogen derived from amino groups also contributes to lipid peroxidation and is a subject of this discussion.

## 1. Introduction

Animal development requires not only cellular proliferation, but also cell death, at the right time and right location in the body. Cell death can be classified into several pathways depending on the types of cells, triggers, and machineries [1,2,3]. Apoptosis is largely involved in development, protection against carcinogenesis, and the pathogenic process. The mechanism of apoptosis is thought to be genetically programmed, as the caspases involved in the cell death process are pre-existing in the form of precursors and, upon proteolytic activation in response to stimuli, solely act on cell death [4,5].

Ferroptosis is an iron-dependent form of nonapoptotic cell death, in which elevation in phospholipid peroxides characteristically induces cell membrane disruption [6]. After the designation of cell death, the elucidation of ferroptosis progressed rapidly, and there are many excellent reviews on this subject, for e.g., refs. [7,8,9,10,11]. The peroxidation of lipids, notably polyunsaturated fatty acids (PUFAs), promotes a radical chain reaction, which progresses in a self-propagating manner. There are now many reports about the molecules that promote ferroptosis by stimulating the production of lipid peroxides, and also the molecules that suppress ferroptosis by scavenging radicals and peroxides or chelating iron [12,13]. Enzymatic inhibitors, as represented by glutathione peroxidase 4 (GPX4), and non-enzymatic compounds, suppress ferroptosis by primarily targeting the lipid peroxidation reaction [14,15].

Ferroptosis is sometimes referred to as programmed necrosis, but is different from other types of programmed cell death in several ways. The production of primary ferroptosis initiators, free iron, and radical electrons, is largely associated with the cellular status concerning malnutrition and abnormal energy metabolism. Lipid peroxidation products have been identified as molecules that directly carry out cell death [8,11,16]; however, so far, all gene products that promote ferroptosis have primary roles other than inducing ferroptosis, and essentially no unique genes dedicated to carrying out ferroptosis have been identified. Thus, ferroptosis appears to occur when oxidative insult is combined with an abnormal nutritional status, genetic defects, or metabolic abnormalities. In this review article, we first provide an overview of this unique type of cell death by referring to the generally accepted processes concerning iron and lipid peroxidation, and then discuss them from the perspective of cellular dynamics and functions, by focusing on the oxidative metabolism of carbon and nitrogen compounds.

## 2. Overview of the Ferroptotic Pathway

Ferroptosis was first characterized in cell death where xCT, a cystine transporter encoded by SLC7A11, was inhibited by erastin under cultured conditions [6]. The inhibition or genetic ablation of xCT decreases the content of free cysteine (Cys) inside cells, leading to a decline in glutathione synthesis [17]. The decrease in glutathione disables GPX4 and lipid peroxidation products accumulate [14,15]. When lipid peroxide accumulates beyond the cellular capacity to maintain the phospholipid bilayer membrane, the membrane eventually ruptures.

### 2.1. Presence of Iron Pushes Lipid Peroxidation beyond the Threshold for Cellular Protection

The elevation in free iron and lipid peroxidation products are unique properties of ferroptosis among the types of cell death, and Fenton chemistry is considered to be tightly associated with ferroptotic cell death [18]. In the Fenton reaction, ferric iron (Fe^3+^), which is in a stored form, is first reduced to ferrous iron (Fe^2+^) by accepting an electron, then the reaction of free ferrous iron with hydrogen peroxide results in the production of hydroxyl radicals (HO•), which are potent oxidants and cause the oxidation of PUFAs, as well as other molecules [19]. When lipid peroxide (LOOH), instead of hydrogen peroxide, is employed, alkoxyl radicals (LO•) are similarly produced. While the reaction of PUFAs with radical species results in alkyl radicals (L•), LOO• are generated by the reaction with oxygen molecules [20]. The reaction of LOO• with PUFAs results in the formation of LOOH and, at the same time, L• is regenerated. The presence of L•, PUFAs, and oxygen molecules continues to generate LOOH as a radical chain reaction, and the presence of free iron further enhances the reaction by generating HO•, additionally.

Many biological reactions produce superoxide (O_2_•) as the first oxygen radical, which is then spontaneously dismutated into hydrogen peroxide. However, O_2_• may donate an electron to other compounds and stimulate the Fenton reaction, if the radical electron is eventually utilized to reduce ferric iron. Superoxide dismutase (SOD), which is encoded by three genes in mammals, accelerates the conversion of O_2_• into hydrogen peroxide, approximately 3000 times faster compared to non-enzymatic dismutation [21]. There is an argument that the SOD-catalyzed reaction is unfavorable because it increases hydrogen peroxide; however, prolonging the lifespan of O_2_• increases the risk of transferring radical electrons to other molecules and, ultimately, to ferric iron. In fact, defects in SOD cause a variety of abnormal phenotypes and pathological conditions in mice, with the genetic ablation of each SOD [22]. The abundance of peroxidases, such as catalase, GPX, and peroxiredoxin (PRDX), rapidly remove hydrogen peroxide, which is considered to be a justification of the physiological significance of SOD. Based on this notion, it is conceivable that, in addition to iron chelation, enzymes and compounds that properly scavenge radical electrons early in the process can inhibit ferroptosis [23,24].

Iron is present mostly in the form of heme and partly as iron–sulfur [Fe-S] clusters and acts in the metabolism of oxygen- and electron-involved reactions [25,26]. Ferritin and transferrin are responsible for iron storage and transport through the vascular system, respectively, but iron bound to these proteins is stable and redox inactive and, hence, does not exhibit harmful effects, including lipid peroxidation, in a general sense. Very little free iron exists, and what is present in a form that can be chelated by high-affinity metal chelators is considered to be a labile iron pool [27]. However, the presence of such iron alone does not trigger lipid peroxidation either. Therefore, how the mobilization of free iron proceeds and where the radical electron is produced to promote lipid peroxidation are crucial issues for understanding the initiation process of ferroptosis.

On the origin of free iron that initiates lipid peroxidation in association with radicals, autophagic degradation of ferritin has been identified as the source, in the early stage of ferroptosis research [28,29]. As proof of this, the inhibition of the lysosomal function prevents the release of free iron into the cytoplasm and the elevation in lysosomal reactive oxygen species (ROS), which eventually associates with the suppression of ferroptosis [30,31,32]. Chaperone-mediated autophagy (CMA) degrades ferritin, which is designated as ferritinophagy, and makes the released iron available for utilization for the corresponding reactions [33]. Nuclear receptor co-activator 4 (NCOA4) is an autophagic receptor for ferritin and plays an essential role in leading ferritin to this degradation machinery [34]. Since the lysosomal membrane becomes a part of the autolysosomal membrane, which is continuously exposed to free iron or other reactive compounds, the lysosomal membrane is uniquely protected against these toxic substances in multiple ways. The contents of α-tocopherol, which suppresses lipid peroxidation by scavenging lipid radicals [35], are rich in the lysosomal membrane, approximately one order of magnitude higher than in other organelle membranes [36]. Upon ferroptotic stimulation, CMA is involved in the degradation of not only ferritin but also GPX4, which may further promote ferroptosis due to its decreased ability to suppress lipid peroxidation [37,38,39]. While treatment with erastin causes ferritinophagy, cell death caused by the inhibition of GPX4 by RSL3 does not appear to require the process [40]. On the contrary, the overexpression of NCOA4 rather delays RSL3-induced cell death, although the reason for this is unclear. These findings also raise an issue concerning the iron source for RSL3-induced cell death. In fact, the sources of iron involved in ferroptosis vary depending on the stimuli, as discussed below.

### 2.2. Elevation in Cell Rupture as a Result of Phospholipid Peroxidation

Since PUFAs are prone to oxidation, numerous studies have been conducted on lipid peroxidation and the resultant cell damage from the perspective of oxidative stress. Regarding lipid peroxidation products, small aldehydic molecules, represented by 4-hydroxy-2-nonenal (HNE), are produced from the degradation of original lipid peroxidation products [41,42]. These small aldehydic molecules can form adducts with nucleotides, proteins, and lipids, which can lead to impaired cell signaling, mutation, or death, when produced in excess [43,44,45]. HNE is also generated from lipid peroxides during ferroptosis, but it is considered that HNE is not a direct executioner of cell death [46]. Instead, it is the peroxidized phospholipids themselves that carry out cell death, and HNE is likely a secondary product of lipid peroxidation [47].

Much information is accumulating about proteins and reactions that promote the peroxidation of phospholipids from perspective of ferroptosis [8,11,16]. PUFAs, represented by arachidonate, are prone to peroxidation. The resulting peroxides react with free iron and further promote the radical chain reaction, which collectively stimulates the accumulation of lipid peroxidation products within the membrane. Some enzymes that are involved in phospholipid synthesis or repair, but not directly in radical formation or the peroxidation reaction, have also been reported to promote ferroptosis [48]. For instance, a long-chain acyl-coenzyme A (CoA) synthetase family member (ACSL) catalyzes the conjugation of fatty acids with CoA. ACSL4, which preferably conjugates arachidonate with CoA, is induced under ferroptotic stimuli and promotes cell death [16,49], but ACSL3, which preferentially converts monounsaturated fatty acids (MUFAs) into acyl-CoA, rather makes cells resistant to ferroptotic stimuli through increasing MUFA-containing phospholipids [50,51]. A recent report implies that phospholipids containing two polyunsaturated fatty acyl tails emerge as the driving force of ferroptosis [52]. These results imply that excessively activated ACSL4 may increase the conjugation of PUFAs to the *sn*-1 position, as well as the *sn*-2 position, and that the resulting phospholipids bearing two PUFAs experience peroxidation and, ultimately, execute ferroptosis.

On the other hand, enzymes that excise fatty acids from phospholipids likely act in the suppression of ferroptosis. Since the majority of the *sn*-2 positions in the glycerol backbone of phospholipids is occupied by PUFAs, phospholipase A2 that removes PUFAs from there may act as a ferroptosis suppressor [53]. Indeed, among the PLA2 family, a type of calcium-independent PLA2 (iPLA2β), encoded by PLA2G6, reportedly suppresses ferroptosis [54,55,56]. This also implies that phospholipid hydroperoxides, but not free peroxidized fatty acids or their derivatives, are responsible for the membrane rupture in ferroptosis. The anti-ferroptotic effects of PLA2 appear to be associated with tumor malignancy. Melanoma is a malignant type of tumor, and the overexpression of PLA2G6 has been reported [57]. The polymorphism of PLA2G6 associates with many malignant tumors, which includes gastrointestinal cancer and melanomas [58]. Although there is no description of the enzymatic activity in these reports, the excessive activation of iPLA2β may make tumor cells resistant to ferroptosis and malignant through removing peroxidized PUFAs. While some types of PLA2s are activated upon stimulation and release arachidonate for producing eicosanoids, iPLA2β are considered to act as a repairing system for damaged phospholipids in order to avoid cell death. Peroxiredoxin 6 (Prdx6) also reportedly acts to repair phospholipid hydroperoxide in a similar manner to GPX4. However, recent reports imply that PRDX6 itself does not have such activity, but regulates GPX4 activity via the selenium synthesis pathway, thereby suppressing ferroptosis [59,60]. 

How P-LOOHs cause cell death remains largely unknown, but recent studies provide clues about the downstream mechanisms of lipid peroxidation. Regarding the mechanism of cell death, there is a report implying that a cation imbalance across the plasma membrane is induced prior to cell rupture [61]. An increase in tension on the plasma membrane is caused by the peroxidation of membrane phospholipids, which leads to the activation of piezo-type mechanosensitive ion channel component 1 (Piezo1) and the transient receptor potential (TRP) channels. As a result, extracellular Na^+^ and Ca^2+^ flow into the cell and, instead, intracellular K^+^ is concomitantly leaked out. On the other hand, lipid peroxidation depresses the activity of Na^+^/K^+^-ATPase, which impedes the restoration of the ionic balance and further promotes osmotic damage [61]. The aberrant action of these ion transporters appears to be responsible for the plasma membrane rupture in the final stage of ferroptosis, but how P-LOOHs act on these ion transport proteins awaits elucidation [62]. Since the properties of the surrounding phospholipids affect the membrane’s protein structure and associated function [63,64], it is possible that P-LOOHs directly affect the ion channel activity through the induction of aberrant protein configuration.

### 2.3. Glutathione/GPX4 System as a Primary Protector against Lipid Peroxidation and Ferroptosis

Glutathione is involved in a variety of reactions, which includes the detoxification of xenobiotics via glutathione conjugation, as well as electron donation for glutathione peroxidase. Two sequential enzymatic reactions on Cys, Glu, and glycine (Gly) by γ-glutamylcysteine synthetase (γ-GCS) and glutathione synthetase (GSS) produce glutathione (Figure 1). Regarding the GPX reaction, a reduced form of glutathione (GSH) donates an electron to the enzymatic reaction of GPX, which results in the reduction of peroxide to alcohol. GPX4 is the dominant enzyme that catalyzes the reduction of phospholipid hydroperoxides, leading to the protection against ferroptosis [65,66,67,68]. The knockout of GPX4 in mice induced embryonic cell death [69]. Both the direct inhibition of GPX4 and the decline in GSH through the inhibition of xCT robustly induces ferroptosis in cultured cells, which indicates that GPX4 is the primary anti-ferroptotic gene. While many other anti-ferroptotic genes have been identified [2,8,11], their protective effects become more pronounced during the inhibition of GPX4 activity. The disabling of GPX4, either through the direct inhibition of the GPX4 enzyme or by decreasing the GSH levels, causes the elevation of phospholipid hydroperoxides and consequent ferroptosis. Like GPX4, a small chemical compound, ferrostatin-1, that suppresses lipid peroxidation is reportedly a potent inhibitor of ferroptosis [70,71].

Among the amino acids building GSH, Cys is less abundant and limits the amount of GSH synthesized by γ-GCS and GSS. Cys can be synthesized through the transsulfuration pathway, coupled with methionine (Met) metabolism in competent cells, such as hepatocytes and certain tumors [72,73]. However, many organs rely on extracellular sources for Cys, plasma Cys, or the oxidized Cys dimer form called cystine, or the degradation products of GSH by γ-glutamyltransferase (GGT) [74]. Cys can be incorporated by cells via the neutral amino acid transporter (NAAT), such as ASCT and system L. However, under cell culture conditions, medium Cys is mostly oxidized to cystine, because cell cultivation is generally performed in atmospheric oxygen (21%), which is approximately four times higher than physiological oxygen (5%). Moreover, xCT is the core transporter protein of system x_c_^−^, which is responsible for the cellular uptake of cystine [17]. The SLC7A11 gene encoding xCT is expressed in limited organs, such as the brain and immune system in healthy animals; however, oxidative stimuli induce the gene’s expression in many organs, most likely through the activation of the transcriptional regulatory protein NRF2 [75,76]. In addition, xCT knockout mice exhibit only mild phenotypic abnormality, namely aberrant plasma redox balance, whereas xCT-deficient fibroblasts cannot be cultivated in conventional culture [77]. Thus, to cope with the oxidative environment, functional xCT is required for most cells under cultivation. For this reason, erastin robustly decreases glutathione by inhibiting xCT and induces ferroptosis in many types of cultured cells, but hardly affects most cells in an in vivo situation. On the other hand, cells with an elevated transsulfuration pathway can also maintain glutathione from endogenously synthesized Cys and are, therefore, resistant to Cys deprivation and xCT inhibition by erastin in cultured cells [78,79].

The depletion of glutathione through the inhibition of γ-GCS by buthionine sulfoximine (BSO) is cytotoxic to certain primary cells and cell lines in cultured conditions [80,81]. However, some cells are less sensitive to the cytotoxic effects of BSO [82]. In fact, the inhibition of γ-GCS by BSO similarly decreases glutathione levels but, instead, increases Cys, which appears to cause the difference between BSO and erastin regarding the ferroptosis-inducing ability [83]. It is also noteworthy to add that erastin inhibits not only xCT but also other proteins, such as mitochondrial voltage-dependent anion channels (VDACs) [84], which may also cause different cellular metabolism. This issue may be associated with a distinctive function of Cys, other than the glutathione component, suppling a sulfur atom to bioactive compounds, which will be discussed in a later section.

Intracellular glutathione is exported via ABC transporters in the form of either GSH, the oxidized form of glutathione (GSSG), or in conjugation with other compounds. Intracellular GSH is also degraded by γ-glutamylcyclotransferase, which is now found to be identical to cation transport regulator protein 1 (CHAC1) and its isoform CHAC2 [85]. CHAC acts in the degradation of intracellular GSH and releases 5-hydroxy proline (5-OP), which is a different product from that of GGT. Oxoprolinase (OPLAH) may convert 5-OP to Glu in an ATP-dependent manner [85]. The meaning behind the fact that GGT, which degrades extracellular GSH, and CHAC, which degrades intracellular GSH, produces different degradation products will also be discussed later. CHAC1 is induced by several stimuli and considered to be a marker for ferroptosis [86]. A mouse study reports that intermittent Met deprivation is beneficial for cancer chemotherapy targeting ferroptosis, which appears to be caused by the stimulated induction of CHAC1 by a short-term deficiency of dietary Met [87]. A Met deficiency decreases Cys production due to a dysfunctional transsulfuration pathway, which causes GSH degradation by means of induced CHAC1 and increases ferroptosis sensitivity. A short-term dietary Met deficiency, however, can still recruit Met from other sources. The replacement of functional CHAC1 with an inactive gene, results in no phenotypic abnormality concerning ferroptosis under conventional breeding conditions, but shows the preservation of skeletal muscle GSH during fasting [88]. This suggests that GSH in skeletal muscle may also act as a peptide pool for gluconeogenesis during emergent malnutrition and fasting conditions.

## 3. Alteration in the Metabolism of Carbon and Nitrogen Compounds upon Ferroptotic Stimulation

The conceptual pathway concerning iron-dependent, non-apoptotic cell death is becoming clear, and some gene products and compounds that either excuse or inhibit the ferroptotic pathway have been discovered [7,8,9,10,11]. However, unlike other types of cell death, many of the gene products that promote ferroptosis are involved in normal biological processes, especially lipid metabolism. Under certain circumstances, upregulation, downregulation, or impairment occurs in these genes, and the resulting products appear to express ferroptosis-promoting effects. However, there remains some ambiguity regarding the initiation of lipid peroxidation reactions; where the radical species and free iron come from, and under what cellular dynamics associate with ferroptosis? To prevent ferroptosis-related disorders, it is essential to gain clarity on these issues. We argue that these issues are important and propose a hypothetical explanation for them in this section.

### 3.1. Association of ROS Production with Cell Proliferation

Researchers have noticed that proliferating cells are more sensitive to ferroptotic stimuli compared to the confluent state under cell culture conditions. Cells in the majority of tissues of adult animals do not divide under healthy conditions, but it is often argued that ferroptosis concerns pathological conditions of tissues, such as inflammation, ischemic damage, and cancer, in which cells awkwardly proliferate [89,90,91]. This suggests that the origins of radical species and iron are associated with properties unique to proliferating cells. Some reports indicate the involvement of cell adhesion molecule integrins in ferroptosis [92,93], but it is unlikely that the dysfunction of integrins is the initial event in the course of all cases of ferroptosis because there is no rational association of their dysfunction with free iron and ROS. Rather, such changes in the functioning of adhesion molecules are considered to be one of the phenomena that occur during the process of cell death.

Figure 2 depicts an outline of the changes in abilities relating to the synthesis of ATP, nucleotides, proteins, and lipids during the cell proliferation cycle. The unique coupling of cyclin/cyclin-dependent kinase (Cdk) regulates the cellular proliferation process, which cycles through four phases: G1, S, G2, and M [94]. Since cells are small immediately after division, they must grow to their original size during the G1 phase via stimulated gene transcription, followed by the translation to proteins and phospholipid synthesis. A hyperfused mitochondrial system with specialized properties is linked to cyclin E buildup for the regulation of G1-to-S progression, which is the period with the maximal activation of cellular metabolism [95]. DNA and mRNA are polymers of (deoxy)ribonucleotide monophosphates and, hence, precursor nucleotide triphosphates are abundantly consumed, not only during DNA replication in the S phase, but also during protein synthesis. Thus, the mitochondrial function concerning energy production and the cell cycle is tightly connected [96].

The G2 phase is well-known for its check point function to reconfirm whether daughter DNA is correctly replicated, and oxidized Cys residues reportedly accumulate in proteins at the M phase [97]. Moderately produced ROS stimulate cell proliferation by modulating phosphorylation signaling through the preservation of receptor signaling of the mitogen [98], but excessively produced ROS suppress mitotic spindle formation through the inhibition of Aurora kinase and results in mitotic arrest [99]. The protein contents represent a balance between their synthesis and degradation, and lysosomes and proteasomes are the major systems responsible for protein homeostasis in cells. In fact, lysosomes are scarcely present in the M phase, presumably due to the preservation of genome integrity against lysosomal nucleases [100,101], which may also lead to the accumulation of oxidized proteins in this specific phase. Accordingly, the accumulation of oxidized proteins during a particular cell cycle phase does not necessarily mean that oxidation has occurred in this particular phase.

In order for the cell to split into two at the end of the M phase, a large amount of phospholipids is required to build the membrane structure for organelles. In fact, free fatty acids and phosphatidylcholine are reportedly synthesized throughout the M phase and are utilized to construct a nuclear envelope, which is required for the completion of the cell division [102]. Thus, lipid synthesis needs to be activated to support the expanding cell size and membranous organelle, both in the G1 phase and G2/M phase. Accordingly, the incorporation of precursor molecules for building blocks and energy metabolism should also be activated to some extent in the G2/M phase as well; although this energy-based metabolic aspect receives little attention compared to DNA synthesis. Regarding lipid metabolism, de novo fatty acid synthesis is required to complete the cell cycle [103]; although the supplementation of palmitate does not rescue the G2/M arrest caused by the pharmacological inhibition of fatty acid synthase [104]. Thus, lipid synthesis is regulated in a complex manner during the cell cycle, and the metabolic reactions require clarification from the standpoint of energetics in order to understand the association of the metabolic changes with ferroptosis.

While cells cultivated in Cys-deprived media undergo ferroptosis, it sounds paradoxical, but the cultivation under the double deficiency of Cys and Met does not induce ferroptosis [78]. Cell cycle arrest characteristically occurs at the end of G1 phase and G2 phase under Met restriction, which is called the Hoffman effect [105]. While Met deprivation causes a decline in the content of its metabolites, *S*-adenosylmethionine (SAM) acts as the methyl group donor during the production of many compounds, such as polyamines, 5′-capping of mRNA, and adrenalin/noradrenalin [106]. Methylation of cytosine at the 5-position occurs in CpG islands as a result of a SAM-dependent mechanism involving the DNA methyltransferase, DNMT1, during DNA replication [39]. DNMT1 gene ablation disables methylation in daughter DNA, and results in cell cycle arrest through hemimethylation in the CpG island [39]. SAM is regenerated from *S*-adenosylhomocysteine by accepting the methyl group from *N*^10^-methyl tetrahydrofolate, a folic acid metabolite related to *N*^5^,*N*^10^-methylene tetrahydrofolate. Since these folic acid metabolites act as co-substrates in the synthesis of deoxy thymidine monophosphate (dTMP), the Met deficiency may also associate with DNA synthesis by cross-talk through folic acid metabolism. The supplementation of SAM alone in regard to the cells that are cultivated under a double deficiency of Cys and Met resumes the cell cycle, but induces ferroptosis, with increased lipid peroxide levels [107]. It appears that cellular metabolism, which is re-activated due to the promotion of the cell cycle, stimulates the production of ROS and lipid peroxides. A recent study reports that cell cycle arrest caused by the activation of p53 enhances the sensitivity to ferroptosis that is induced by GPX4 inhibitors, but not by xCT inhibitors [108]. In this study, p53 was activated by the inhibition of the ubiquitin ligase, MDM2, by nutlin-3, so that the cells were generally arrested in the G1/S phase, where energy metabolism in the mitochondria may still be active and, hence, produces an abundance of ROS. Taking the observations by Gryzik et al. [40] and Rodencal et al. [108] together, ferritinophagy appears to be involved in erastin-induced ferroptosis, but mitochondria-derived iron/radicals may be responsible for ferroptosis caused by GPX4 inhibition. This also implies the significance of the mitochondria-specific form of GPX4, which is translated from alternatively spliced mRNA [109], in ferroptosis caused by GPX4 inhibition.

Nutrient starvation often leads to tumor cell death. Iron delivery via transferrin is generally essential for cell growth, but, in cultured cancer cells under amino acid deprivation, iron that is released within the secondary endosome is responsible for triggering ferroptosis [110]. Moreover, glutamine (Gln) is a vital amino acid that supports tumor cell survival and growth through a process called glutaminolysis, which is mediated partly by providing the building blocks for nucleic acid synthesis and, also, after conversion to glutamate (Glu), for glutathione [111]. Surprisingly, glutaminolysis also induces ferroptosis in amino acid-deprived cancer cells, which appears to be associated with the ROS released during elevated Glu metabolism [110,112]. There are other cases in which Glu hypermetabolism causes ferroptosis. Some γ-glutamyl peptides are produced by the intrinsic activity of γ-GCS under Cys deprivation conditions [113,114]. Ophthalmate, a mimetic of GSH, is a γ-glutamyl peptide abundantly produced in liver treated with acetaminophen, a popular analgesic drug [115]. Because acetaminophen reportedly induces ferroptosis, along with glutathione deprivation [116,117], it is considered that ophthalmate is thus produced under the condition of glutathione consumption, due to the conjugation reaction of acetaminophen and Cys insufficiency that may play a certain role. However, no physiological significance in terms of the production of ophthalmate itself has been known. In the meantime, Kang et al. [118] have reported on a potential role of γ-glutamyl peptide production by γ-GCS, per se, in the protection against ferroptosis from the metabolic aspect of Glu. When Glu is consumed for the production of γ-glutamyl peptides under Cys deprivation, the catabolism of Glu-derived 2-oxoglutarate (2-OG) in the tricarboxylic acid (TCA) cycle is attenuated, which decreases the production of oxygen radicals in the respiratory chain and eventually inhibits Cys deprivation-induced ferroptosis (Figure 3). The production of γ-glutamyl peptides is elevated not only by acetaminophen overdose, but also in some hepatic injuries in humans [119], which implies that ferroptosis may also be involved in such diseases and that an elevation in γ-glutamyl peptides is a hallmark of ferroptosis in the liver.

Moreover, the association of Glu metabolism with ferroptosis may also provide a clue to understanding why extracellular GSH and intracellular GSH experience different catabolic reactions with each other. While GGT hydrolyzes extracellular GSH to produce Glu and cysteinyl–glycine in the absence of sufficient acceptor molecules, the degradation of intracellular GSH by CHAC produces 5-OP, which may be secondarily converted into Glu by OPLAH in an ATP-dependent manner [74,85]. CHAC1 is reportedly induced by Cys deficiency through xCT inhibition and is proposed to be a maker of ferroptosis [86]. If Glu is produced to a great extent through GSH degradation inside cells, the catabolic reaction increases the release of radical electrons, which may promote lipid peroxidation and eventual ferroptosis, as discussed. Based on this assumption, it is conceivable that the production of 5-OP by CHAC, instead of Glu, plays a buffering role in order to attenuate the Glu flux in regard to the catabolic process, although the excessive activation of CHAC1 markedly declines GSH and promotes ferroptosis through disabling GPX4 [87,120].

### 3.2. Iron Release in Association with Cellular Metabolism

Ferroptosis is effectively inhibited by iron chelation, indicating that iron release occurs before or simultaneously with lipid peroxidation and plays central roles in executing ferroptosis [6]. Ferritinophagy is a reliable iron source upon erastin treatment, but it is not clear where iron comes from in other situations, notably in the case of ferroptosis induced by GPX4 inhibition [40]. While ferritinophagy may recruit free iron to induce the lipid peroxidation reaction in the cytoplasm under certain circumstances, mitochondria are also organelles rich in iron and produce an abundance of radical electrons, especially under activated energy catabolism. In fact, the structural destruction of mitochondria is one of hallmarks of ferroptosis [6]. While the essential roles of mitochondrial ROS and/or iron are implied for the execution of ferroptosis [6,112], a report shows that the deprivation of mitochondria does not affect ferroptosis induced by several stimuli [121]. This issue remains unclear, but subsequent studies largely support a mitochondrial origin of the relevant radical species depending on the stimulus [8,11,122].

Cells with GPX4 inhibition typically induce ferroptosis, which implies that certain levels of lipid peroxidation constantly occur and that GPX4 keeps removing them to make cells viable. Iron and copper are micronutrient minerals and mostly act in a redox reaction in the protein-bound form but, once released, they indiscriminately catalyze the formation of oxygen radicals in the presence of peroxides, mostly hydrogen peroxide, and trigger unfavorable reactions [123]. Most iron binds porphyrin to form heme, which is in a rigid structure and is not redox reactive by itself. An iron–sulfur cluster [Fe-S] is another form of iron-containing redox cofactor that is produced in mitochondria and is abundantly present in the electron transport complex (ETC) and some other enzymes, including aconitase encoded by ACO2 and succinate dehydrogenase (SDH) in the TCA cycle [124,125]. Few iron ions are directly coordinated to proteins to form the catalytic center of enzymes, such as lipoxygenase [126,127]. While the [Fe-S] cluster is often utilized in electron-transfer reactions, some of them work to maintain the protein structure [128]. From the perspective of iron homeostasis, it is necessary to consider the possible contribution of the iron-regulatory protein, also called cytosolic aconitase 1 (ACO1). ACO1 contains a [4Fe-4S] cluster that is highly susceptible to the cellular iron status. One iron in the [4Fe-4S] cluster of ACO1 is not coordinated with Cys-SH, which enables the iron status in the cytoplasm to be sensed. Under iron-insufficient conditions, one iron atom is released from the [4Fe-4S] cluster, which makes ACO1 bind the iron-responsive element (IRE) in the mRNA encoding iron-metabolizing proteins, represented by the ferritin and transferrin receptor. When the [3Fe-4S] form of ACO1 binds IRE localized in the 5′-untranslated region of the ferritin mRNA, its translation is inhibited. In the meantime, its binding to IRE is localized in the 3′-untranslated region of the transferrin receptor mRNA, which increases the transferrin receptor protein through the protection of mRNA from degradation. As a result, in both cases, more iron becomes available for the required reactions.

Mitochondria are central organelles that metabolize iron, as well as oxygen, which includes synthesizing heme and the [Fe-S] cluster. Iron, which is transferred from cytosol, is directly utilized to synthesize the iron complex or is transiently stored in mitochondrial ferritin, which has a high level of structural similarity with cytosolic H-ferritin [129]. Erastin-induced ferroptosis and ischemia/reperfusion-induced cerebral damage are alleviated by the overexpression of mitochondrial ferritin, which appears to be due to decreasing free iron toxicity in lipid peroxidation [130,131]. Thus, mitochondrial ferritin likely plays a role in iron homeostasis in a similar manner to cytosolic ferritin, but information on this molecule is limited and further clarification of its role in mitochondria-associated ferroptosis is needed.

Since Fe-containing enzymes are abundantly present in mitochondria, ROS/reactive nitrogen oxide species (RNOS) are unequivocally produced and potentially a target of the TCA cycle and ETC [132,133]. Among the enzymes in the TCA cycle, ACO2 is the most sensitive target of several ROS/RNOS because the [4Fe-4S] cluster required for the aconitase activity is highly susceptible to oxidative insult due to the unstable nature of its coordination with protein molecules [21,134,135]. Enzymes in the TCA cycle act to perform degrading acetyl-CoA under physiological conditions. However, under pathological conditions, notably inflammation, ROS are largely produced because electrons may be leaked out during the electron-transfer reaction of ETC [136,137]. The resulting ROS may oxidatively damage susceptible enzymes and lipids. ACO2 is also proposed to act as an iron-sensing regulator in mitochondria, in addition to its established role in the catalysis involving the conversion of citrate to *iso*-citrate. Mitochondrial ACO2 is reportedly inactivated by the reaction with ROS and RNOS, which appears to be attributed to the release of ferrous iron [134,138]. The remodeling of carbon metabolism by the TCA cycle has been established in macrophages, which experience polarization into the active form, designated as the M1 type [135,139]. Three [Fe-S] clusters, [2Fe-2S], [3Fe-4S], and [4Fe-4S], are present in succinate dehydrogenase, which is sensitive to inactivation by superoxide and NO, as well as itaconate, a metabolite produced via the broken TCA cycle of M1 macrophages [139,140,141]. Although direct evidence is not provided, the destruction of the [Fe-S] cluster of SDH may also occur, as observed in aconitase, and may release free iron to trigger ferroptosis.

Compared to the role of TCA cycle remodeling in macrophage polarization, essentially no attention has been paid so far to the iron released during this process. Macrophages are known to be resistant to the toxic action of iron, probably due to unique iron exclusion systems, which includes the lysosomal iron pump protein natural resistance-associated macrophage protein 1 (Nramp1) and the iron export protein ferroportin [142]. A lack of such iron-detoxification systems may render ordinary cells susceptible to iron toxicity and cause ferroptosis in non-immune cells. Thus, it is conceivable that, in association with the electrons leaked from ETC, the released iron from [Fe-S] clusters may be responsible for lipid peroxidation in the mitochondrial membrane. Consistent with this notion, exposure to hyperoxia reportedly causes preferential degradation of [Fe-S] cluster-containing proteins in the ETC of mitochondria in the lungs of mice [143].

Since the ACO2 protein is abundantly present and susceptible to oxidation, it is speculated that the iron released from the [Fe-S] cluster plays a role in ferroptosis. In fact, the [Fe-S] cluster assembly protein, ISCU, suppresses the ferroptosis that is induced by dihydroartemisinin through the regulation of the mitochondrial iron status [28]. Cysteine desulfurase, NFS1, that recruits sulfur from Cys to ISCU to assemble the [Fe-S] cluster also suppresses ferroptosis in lung tumor cells [144]. Moreover, frataxin, which modulates the assembly of [Fe-S] clusters, forms the cysteine desulfurase complex and suppresses ferroptosis [145]. We do not have direct evidence, but it can be speculated that ROS originated from the ETC attack [4Fe-4S] cluster in ACO2 and other susceptible enzymes, as observed in macrophage, which results in the release of ferrous iron, together with hydrogen peroxide, and collectively induces ferroptosis. The possible involvement of the [Fe-S] cluster in iron homeostasis may also be supported by studies on newly found [Fe-S] containing proteins. Proteins with the CDGSH iron sulfur domain (CISD) contain the [2Fe–2S] cluster and appear to be responsible for the [2Fe–2S] cluster relay between the mitochondria and other cellular components [25,146]. From the perspective of ferroptosis, CISD proteins act differentially; CISD1 and CISD3 stimulate ferroptosis, but CIDS2 confers resistance against ferroptosis. These inconsistent contributions to ferroptosis may be attributed to the unique function of individual proteins in the assembly of the [Fe-S] cluster.

### 3.3. Nitrogen Metabolism-Associated ROS Formation

While amino acids may be utilized as a carbon source through the oxidatively remodeled TCA cycle, the consumption of amino acids produces byproducts containing nitrogen. While Glu is the most abundant amino acid in cytosol, it is also largely formed from 2-OG by reactions involving aminotransferase, unique to the corresponding amino acids. Oxidative cleavage of Glu by glutamate dehydrogenase results in the production of 2-OG and ammonia. Liver and a part of intestinal cells synthesize urea from ammonia and carbon dioxide via the urea cycle. However, other cells mostly lack the expression of genes for carbamoyl phosphate synthetase I and ornithine transcarbamylase (OTC) [147], so they cannot detoxify ammonia via the urea cycle. Nevertheless, the expression of arginase enables the hydrolyzation of arginine (Arg) to ornithine and urea in many cells. Arg may come from extracellular milieu or be generated through the incomplete urea cycle (Figure 4). Citrulline is conjugated with aspartate (Asp) by argininosuccinate synthetase 1 (ASS1) to generate argininosuccinate, which is in turn cleaved to Arg and fumarate by means of argininosuccinate lyase (ASL). In the absence of OTC, citrulline is not generated from ornithine, but Arg is converted into nitric oxide (NO) and citrulline by the reaction of nitric oxide synthase (NOS). Thus, citrulline can be regenerated from Arg, so Arg is the compound located at the branching point of urea cycle-related metabolic pathways. In the meantime, Asp is effectively regenerated by the reaction of aspartate aminotransferase (AST) that transfers the amino group of Glu to oxaloacetate [148]. Given these collective reactions, nitrogen in the form of the amino group of many amino acids can be converted into either urea or NO via the coordinated action of these metabolizing pathways. The conjugation of Asp with citrulline by ASS1 promotes the conversion of Glu and oxaloacetate to 2-OG and Asp, respectively, by AST. Accordingly, ASS1 decreases mitochondrial ROS production through the consumption of 2-OG by means of the reductive pathway and, consequently, suppresses erastin-induced ferroptosis in non-small-cell lung cancer cells [149]. In the meantime, 2-OG is reversed to form citrate in the TCA cycle, which is then recruited to synthesize palmitic acid, followed by palmitoleic acid, a mono-unsaturated fatty acid, which also tends to suppress ferroptosis [50].

The absence of OTC, however, increases ornithine. Under some circumstances, ornithine decarboxylase (ODC) is induced and catalyzes the conversion of ornithine to putrescine, the first polyamine species produced in cells [150]. The resulting putrescine is further converted into spermidine by spermidine synthase, and then into spermine by spermine synthase, using SAM as a co-substrate. Polyamines exhibit a variety of physiological activities, including the stimulation of cell proliferation and the exertion of anti-inflammatory reactions [150,151,152,153]. Regarding anti-ferroptotic action, spermidine maintains the expression of xCT and GPX4 through the inhibition of ferritinophagy and improves rheumatoid arthritis that is exacerbated by interleukin-1β-mediated cell death [154]. Spermidine also modulates ferroptosis and exerts beneficial action in regard to Alzheimer’s disease [155].

On the other hand, enzymes catabolizing polyamines produce ROS as byproducts that may cause oxidative damage to cells and may be associated with diseases [156,157]. Spermine oxidase (SMOX) oxidizes spermine to spermidine, aminoaldehyde, and hydrogen peroxide. Spermine/spermidine *N*^1^-acetyltransferase (SAT1) catalyzes, transferring the acetyl group to spermine and spermidine from acetyl-CoA, resulting in *N*^1^-acetylspermine and *N*^1^-acetylspermidine, respectively. Polyamine oxidase (PAO) converts *N*^1^-acetylspermine and *N*^1^-acetylspermidine back to spermine and spermidine, respectively, and produces hydrogen peroxide as a byproduct. While PAO is constitutively expressed, SAT1 is highly inducible by various stimuli and limits polyamine metabolism [150], so that SAT1 induced in response to some stimuli may promote the PAO-mediated production of hydrogen peroxide. Accordingly, exogenously administered polyamines suppress ferroptosis under some pathological conditions [55,158], but ROS may be produced during their metabolism and participate in ferroptosis execution, depending on the expression of SAT1. The activation of p53 induces SAT1, which aggravates ferroptosis [159]. The relief of neuropathic pain symptoms by electroacupuncture appears to be associated with ferroptosis in the dorsal root ganglion, in which SAT1 and arachidonate-specific lipoxygenase 15 (ALOX15) are involved [160]. The combined administration of gemcitabine and cisplatin upregulates SAT1 in pancreatic ductal adenocarcinoma via the inhibition of transcriptional factor Sp1, leading to the induction of ferroptosis in the tumor cells [161]. Arginine promotes polyamine synthesis through increasing ornithine by the action of arginase, so that arginine indirectly increases ferroptosis sensitivity in cancer cells [162]. On the contrary, citrulline reportedly suppresses ferroptosis in iron-overload thymus, which appears to be attributable to the direct inhibition of ferritinophagy [163]. Thus, compounds produced by nitrogen metabolism also have different effects on ferroptosis, but the relationship between the polyamine pathway coupled with the urea cycle and ferroptosis varies depending on cellular and environmental factors and cannot be definitively determined.

### 3.4. Ferroptosis Suppression by the Redox System

Since oxygen radicals initiate the lipid peroxidation reaction and lead to ferroptosis, scavenging radical species, by either enzymatic or non-enzymatic compounds, is the fundamental form of prevention against ferroptosis-related diseases [48]. Tocopherol, also called vitamin E, is the primary scavenger of the lipid radical (L•) and, in association with other redox systems, terminates the chain reaction and suppresses the further production of P-LOOH [35]. The tocopherol radical (Toc•) may be reduced by vitamin C (ascorbate), as a reducing agent [164]. Since vitamin C can donate an electron to iron ions, it rather increases ROS production and may promote death in cells with high free iron content, as typically observed in cancer cells [165]. Thus, whether vitamin C exerts beneficial or detrimental effects is highly dependent on environmental elements, notably iron. The presence of free iron, however, may rather enhance the production of coenzyme Q (CoQ), which is synthesized in mitochondria and is involved in the electron-transfer reaction between ETCs. CoQ is also partly transferred to the plasma membrane via STARD7 and reportedly protects membrane phospholipids from peroxidation [166]. CoQ oxidoreductase reduces CoQ in a NADPH-dependent manner, leading to the suppression of ferroptosis independently from GPX4 [167,168]. According to this function, CoQ oxidoreductase is now renamed as the ferroptosis-suppressor protein 1 (FSP1). Vitamin K donates an electron and is converted into a vitamin K radical, which results in the suppression of lipid peroxidation via its non-canonical function [169]. FSP1 again reduces the vitamin K radical and results in the suppression of ferroptosis. GTP cyclohydrolase-1 (GCH1) produces tetrahydrobiopterin, which indirectly maintains the CoQ redox state and also suppresses ferroptosis [170].

Quite recently, three groups have reported the anti-ferroptotic effects of 7-dehydrochoresterol (7DHC) through endogenous radical scavenging action [171,172,173]. While 7DHC and the hydrocarbon tail of CoQ are produced through the cholesterol-synthesizing pathway, oxidative stress likely promotes cell death on one hand, but stimulates the pathway acting in a protective capacity, on the other hand. Since the activity of the cholesterol-synthesizing pathway varies greatly depending on the cells and tissues involved, the contribution of these compounds in suppressing ferroptosis also appears to vary by cell type.

In order to preserve the protective action, most anti-ferroptotic systems, including GSH-GPX4 glutathione reductase and CoQ/vitamin K-FSP1, are coupled with corresponding reductase enzymes that commonly require NADPH as the primary electron donor. NADPH is largely produced via the pentose phosphate pathway, notably glucose-6-phosphate dehydrogenase (G6PDH), while the decarboxylation of malate to pyruvate by the malic enzyme also produces some NADPH. Therefore, the enhanced expression of enzymes in the pentose phosphate pathway could indirectly contribute to the suppression against ferroptosis [174]. The expression of some enzymes constituting the pentose phosphate pathway is robustly regulated by NRF2, which is activated by oxidative stress and alkylating agents [175,176]. Thus, NRF2 works to suppress ferroptosis at the level of the most basic reaction.

## 4. ROS from Other Enzymatic Sources

Given the consumption of most oxygen in mitochondria, it is easy to understand that mitochondria are one of the promising sources of ROSs that cause ferroptosis. However, there are many other enzymes that catalyze reactions using oxygen, implying that these enzymes may also act as a source of ROS. Here, we will discuss well-known examples of the production of ROS and the origin of iron. To date, only a few oxidase/oxygenase enzymes have been shown to be directly associated with ferroptosis, which may be due to the need for free iron to promote radical chain reactions above the threshold levels of lipid peroxides. Some enzymes using oxygen molecules, notably arachidonate-specific lipoxygenase (ALOX) and P450 reductase (POR), reportedly produce lipid peroxidation products that occasionally execute membrane destruction and cell death [11,177,178]. These processes appear to be independent from the central metabolic reactions described above.

### 4.1. Arachidonate-Specific Lipoxygenase (ALOX)

Lipoxygenase (LOX), notably arachidonate-specific isozymes, ALOX, is deeply involved in ferroptosis [177,178,179]. While ALOX and cyclooxygenase (COX) are enzyme families that employ iron as a cofactor to oxidize unsaturated fatty acids, mostly arachidonate, and produce LOOH as intermediary compounds. Regarding ferroptosis, there are many reports on the involvement of ALOX in lipid peroxidation, but very few on that of COX. This appears to be associated with the iron status in these enzymes, i.e., ALOX contains non-heme iron, but COX contains heme iron [48]. ALOX5 is required for ferroptosis upon treatment with some stimulants but is not required for erastin-induced ferroptosis [180]. Non-heme iron coordinated with the ALOX protein supports the peroxidation reaction, but heme plays a role in the COX reaction. Moreover, 5-LOX is highly unstable compared to COX and is readily inactivated in the presence of oxygen [127]. This LOX inactivation occurs even during normal enzymatic process, which is reportedly based on the mechanism of suicide inactivation [181]. Rabbit ALOX15 contains a six-coordinate ferrous iron [126], and during an enzymatic reaction, the inactivation of the enzymatic activity occurs [127,181]. This was found to be caused by covalent modification, as demonstrated in the experiment using an intermediary compound 15*S*-hydroperoxy-5,8,11,13-eicosatetraenoic acid (15-HETE) [182]. While macrophage is known to be less sensitive to the ferroptotic stimuli, NO, which is largely produced in activated macrophages, it appears to cause the nitroxygenation of 15-hydroperoxy-eicosa-tetra-enoylphosphatidylethanolamine and suppress ferroptosis [183]. Moreover, 5-LOX is also inactivated by hydrogen peroxide upon incubation with oxygen via a non-turnover mechanism [184]. There is no description of what happens to the iron bound to 5-LOX, so it is unclear whether the iron is actually released from the protein or not. Tyrosinase, which catalyzes tyrosine oxidation during the first step of melanin synthesis, coordinates two copper ions in its active center. When rhododendrol, a skin-whitening ingredient, is excessively metabolized by tyrosinase, melanocytes undergo cell death, which is likely due to the action of a suicide substrate-dependent mechanism [185]. Since a copper-chelator D-penicillamine alleviates melanocyte death, the released copper likely generates hydroxyl radicals based on the Fenton chemistry-like mechanism, so a molecular mechanism similar to ferroptosis may be involved in this case as well. In the case of COX, superoxide produced during the oxygenase reaction is associated with lipid peroxidation [186], whereas heme is a stable compound and is unlikely to be released during the course of the enzymatic reaction. This may be the reason why COX-induced lipid peroxidation does not accumulate above the threshold to induce ferroptosis, although both LOX and COX are similarly involved in the peroxidation of arachidonic acid.

### 4.2. P450 Oxidoreductase (POR)

Cytochrome P450 (CYP) acts in the first phase of detoxification reactions of xenobiotics and introduces an oxygen-containing group, making xenobiotics susceptible to being conjugated. Cytochrome P450 reductase (POR) transfers electrons from NADPH to CYP through endogenous FAD and FMN in the enzyme [187], which enables the continuous oxygenation of substrate compounds. However, superoxide may also be produced during this electron-transfer reaction. Since this type of detoxification reaction is dominantly active in the liver and kidney, CYP-mediated ROS production is largely associated with hepatic damage caused by drugs and xenobiotics. While POR, which transfers electrons from NADPH to CYP, results in the production of hydrogen peroxide under certain circumstances, it may cause lipid peroxidation to execute ferroptosis [188]. Since acetaminophen first undergoes oxygenation by CYP, followed by glutathione conjugation, lipid peroxidation could be induced by the POR/CYP system and trigger ferroptosis [116,117]. These oxygenase enzymes contain heme, and it is unlikely that iron is released from the enzymes during the reaction, as discussed in regard to lipid peroxidation by LOX vs. COX. The potential involvement of heme iron has been proposed [188], but the details in regard to the mechanism are not clear. A subsequent study also implies the existence of a free iron-dependent mechanism for POR-involved ferroptosis [189], although it is still unclear where free iron comes from. Since heme oxygenase decomposes heme and releases iron, as well as carbon monoxide, some studies imply the involvement of this enzyme in releasing free iron from heme and consequent ferroptosis [190]. These findings imply coordinated action of the POR/CYP system and heme oxygenase in inducing ferroptosis; however, further investigation is needed to determine how heme iron is recruited to the lipid peroxidation reaction.

### 4.3. Potential Involvement of Other Oxidase/Oxygenase

ROS are intentionally generated by NADPH-dependent oxidases (NOX) and act for the purpose of anti-microorganisms in leukocytes, for instance. The generated ROS are released into autophagosomes, which are formed when microorganisms are phagocytosed. GKT137831, a NOX1/4-specific inhibitor, first reportedly suppresses erastin-induced ferroptosis, and inhibits NADPH production through the pentose phosphate pathway, which partly prevents ferroptosis too [6]. Among the NOX family members, NOX4 is expressed in non-immune cells and generally modulates the cellular signaling process rather than the anti-microorganism. The activation of NOX4 promotes ferroptosis of astrocytes via the impairment of mitochondrial metabolism and the production of lipid peroxides, which may be associated with Alzheimer’s disease [191]. However, it is unclear how NOX4 specifically affects mitochondrial function. NOX4 is overexpressed in glioma and may be linked with a worse prognosis [192]. Other oxidase/oxygenase are also known to generate radical species and lead to lipid peroxidation and, hence, the promotion of ferroptosis by such enzymes may emerge in future studies.

### 4.4. Iron-Independent Lipid Peroxidation Induced Cell Death (Lipoxytosis) Regulated by GPX4

GPX4 can reduce phospholipid hydroperoxide and suppress iron-dependent lipid peroxidation-induced cell death, ferroptosis. Recently, Tsuruta et al. [193] reported that the depletion of GPX4 in Tamoxifen-inducible GPX4-deficient mouse embryonic fibroblast cells (ETK1 cells) induced iron-independent lipid peroxidation induced slowly progressive cell death, designated as lipoxytosis. In ETK1 cells, Tamoxifen-inducible gene disruption of GPX4 induces slow cell death at ~72 h. In contrast, RSL3- or erastin-induced ferroptosis occurred quickly, within 24 h. Therefore, Tsuruta et al. [193] investigated the differences in these mechanisms between GPX4 gene disruption-induced cell death and RSL3- or erastin-induced ferroptosis. A GPX4 deficiency induced lipid peroxidation at 24 h in Tamoxifen-treated ETK1 cells, which was not suppressed by iron chelators; although lipid peroxidation in RSL3- or erastin-treated cells induced ferroptosis that was inhibited by iron chelators. Although MEK1/2 inhibitors suppressed both GPX4-deficient cell death and RSL3- or erastin-induced ferroptosis, GPX4-deficient cell death was MEK1-dependent, but RSL3- or erastin-induced ferroptosis was not. In GPX4-deficient cell death, the phosphorylation of MEK1/2 and ERK2 was observed 39 h after lipid peroxidation, but ERK1 was not phosphorylated. Selective inhibitors of ERK2 inhibited GPX4-deficient cell death. However, ferroptosis induced by RSL3 and erastin was not inhibited by ERK1/2 inhibitors or ERK1 and ERK2 selective inhibitors. These findings suggest that iron-independent lipid peroxidation due to GPX4 disruption induced cell death via the activation of MEK1/ERK2 as a downstream signal of lipid peroxidation in Tamoxifen-treated ETK1 cells. On the other hand, although we cannot exclude the possibility that the inhibitory effect of MEK1/2 inhibitors on ferroptosis induced by RSL3 or erastin is mediated solely by MEK2, it is possible that the inhibition of MAPKK, other than MEK, may inhibit RSL3- or erastin-induced ferroptosis These results indicates that GPX4 gene disruption induces slow cell death, i.e., lipoxytosis, and involves a different pathway from RSL3- and erastin-induced ferroptosis in ETK1 cells.

## 5. Concluding Remarks

Ferroptosis plays a pivotal role in the development of a variety of pathological conditions, including inflammation, ischemic diseases, neurodegenerative diseases, and cancer. The accumulating data indicate that metabolically active cells are susceptible to ferroptosis, suggesting that the radicals generated in active metabolism are responsible for lipid peroxidation. These metabolic pathways include not only central carbon metabolism, but also nitrogen metabolism. The identification of the source of the causative radical species could contribute to the development of effective therapeutics for ferroptosis-related diseases.

## Figures and Tables

**Figure 1 ijms-25-07544-f001:**
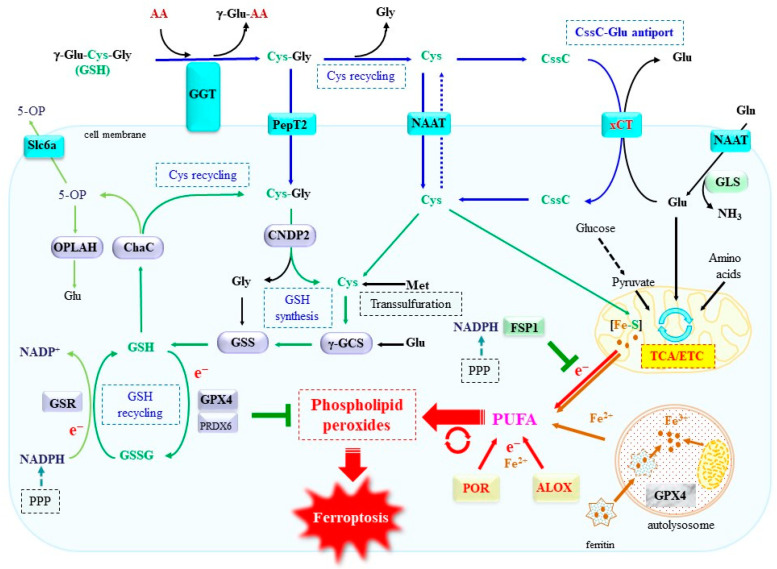
Principle pathways for induction of and protection from ferroptosis. Free iron mainly comes from either ferritinophagy or mitochondrial iron, but also originates from iron-containing enzymes. These components may also become the source for radical electrons to initiate lipid peroxidation. Cys availability determines the cellular levels of glutathione, which supports GPX4 in the protection against lipid peroxidation. Part of the central metabolic pathways discussed in the text are also depicted. AA, amino acid; γ-Glu-AA, γ-glutamyl amino acid; CssC, cystine; PepT2, dipeptide transporter; PPP, pentose phosphate pathway; 5-OP, 5-hydroxy proline; Slc6a, 5-oxoproline transporter; TCA, tricarboxylic acid cycle; ETC, electron transport chain; GSR, glutathione reductase; GLS, glutaminase; FSP1, ferroptosis-suppressor protein 1; POR, cytochrome P450 reductase; ALOX, arachidonate-specific lipoxygenase.

**Figure 2 ijms-25-07544-f002:**
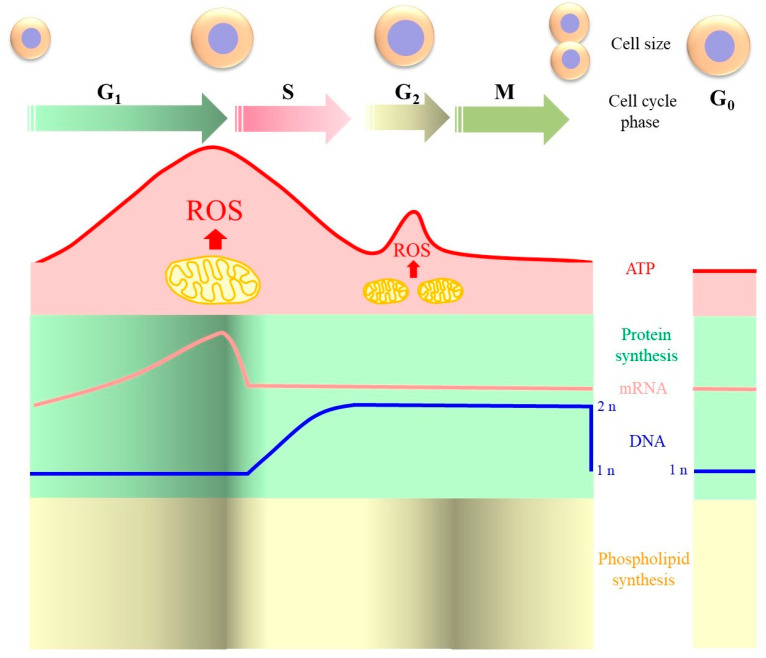
Relationship between cell cycle, energy metabolism, and polymer synthesis. This diagram schematically shows the relationship between events that occur in each phase of the cell cycle and the metabolism that synthesizes biopolymers. Darker colored areas mean more active in terms of metabolism.

**Figure 3 ijms-25-07544-f003:**
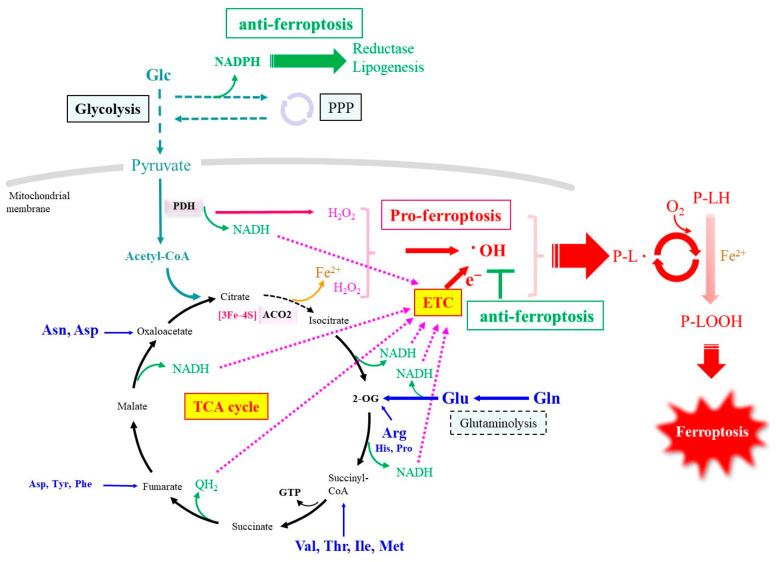
Radical production pathway involved in ferroptosis centered on the TCA cycle. This diagram shows the process in which carbon derived from glucose and carbon backbones of amino acids are metabolized during the TCA cycle, and how the electrons generated during this process are involved in ferroptosis. Only a few key enzymes that play important roles in metabolism are shown. PPP, pentose phosphate pathway; P-L•, phospholipid radical; PDH, pyruvate dehydrogenase.

**Figure 4 ijms-25-07544-f004:**
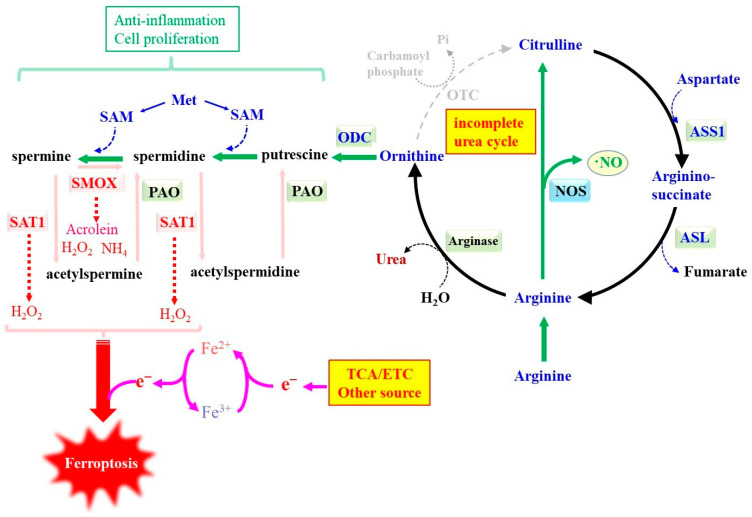
A potential role of the polyamine pathway in ferroptosis. Polyamine synthesis that occurs in conjunction with an incomplete urea cycle associates with the production of ROS responsible for ferroptosis. OTC deficiency activates polyamine synthesis by accumulating ornithine. The metabolic process of polyamines generates hydrogen peroxide, which may promote lipid peroxidation and induce ferroptosis.

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
