# Peer review of "Oxidative Metabolism as a Cause of Lipid Peroxidation in the Execution of Ferroptosis"

_ijms, 2024, doi:10.3390/ijms25147544_

Round 1

Reviewer 1 Report

Comments and Suggestions for Authors

This is a very interesting review paper which relies on the multitude of effects involved with ferroptosis and its relationship with oxidative processes and lipid peroxidation.  Authors conclude that ferroptosis plays central roles in a variety of pathological conditions, including inflammation, ischemic diseases, neurodegenerative diseases, and cancer. They further suggest that active metabolism are responsible for lipid peroxidation including not only central carbon metabolism but also nitrogen metabolism.

 All illustrations are clear and easy to follow.

 I have only few suggestions that could improve the current manuscript.

 This manuscript is overall well written, but a careful revision for grammar and style will benefit the entire paper. 

 Experimental data involving execution of ferroptosis in different diseases or functional status “described over the text” could be arranged in a table to clearly indicate the molecular response, animal or in vitro model, and the dependence of ferroptosis-related mechanisms.

 Minor comments

 Correct “disumtation” in the line 88

 Correct “whjch” in the line 376

 Correct “G6PD” to “G6PDH” in line 611

 Correct “ferropsosis” in the line 693

Comments on the Quality of English Language

This is a very interesting review paper which relies on the multitude of effects involved with ferroptosis and its relationship with oxidative processes and lipid peroxidation.  Authors conclude that ferroptosis plays central roles in a variety of pathological conditions, including inflammation, ischemic diseases, neurodegenerative diseases, and cancer. They further suggest that active metabolism are responsible for lipid peroxidation including not only central carbon metabolism but also nitrogen metabolism.

 All illustrations are clear and easy to follow.

 I have only few suggestions that could improve the current manuscript.

 This manuscript is overall well written, but a careful revision for grammar and style will benefit the entire paper. 

 Experimental data involving execution of ferroptosis in different diseases or functional status “described over the text” could be arranged in a table to clearly indicate the molecular response, animal or in vitro model, and the dependence of ferroptosis-related mechanisms.

 Minor comments

 Correct “disumtation” in the line 88

 Correct “whjch” in the line 376

 Correct “G6PD” to “G6PDH” in line 611

 Correct “ferropsosis” in the line 693

Author Response

Thank you very much for your efforts on reviewing and criticism on our manuscript. We are grateful to the reviewers for their valuable comments that helped improve the manuscript. Our individual responses follow after their comments.

Reviewer 1

This is a very interesting review paper which relies on the multitude of effects involved with ferroptosis and its relationship with oxidative processes and lipid peroxidation.  Authors conclude that ferroptosis plays central roles in a variety of pathological conditions, including inflammation, ischemic diseases, neurodegenerative diseases, and cancer. They further suggest that active metabolism are responsible for lipid peroxidation including not only central carbon metabolism but also nitrogen metabolism.

 All illustrations are clear and easy to follow.

 I have only few suggestions that could improve the current manuscript.

 This manuscript is overall well written, but a careful revision for grammar and style will benefit the entire paper. 

 Experimental data involving execution of ferroptosis in different diseases or functional status “described over the text” could be arranged in a table to clearly indicate the molecular response, animal or in vitro model, and the dependence of ferroptosis-related mechanisms.

Our responses: Thank you for kind advice. This review is not specific to any particular disease but rather discusses the molecular mechanisms underlying ferroptosis, a novel form of cell death, from a metabolic perspective that has received little attention so far. If this review is focusing on a specific organ or disease, the reviewer’s suggestions would be useful for readers. However, because the diseases covered in this article are diverged and many different model systems are employed in each study, it is extremely difficult to summarize them in tables. We believe that there are cases where summarizing the contents in tables does not always help readers’ understanding, as in this manuscript.

 Minor comments

 Correct “disumtation” in the line 88

Our responses: Thank you very much for pointing out. We have corrected it.

 Correct “whjch” in the line 376

Our responses: Thank you very much for pointing out. We have corrected it.

 Correct “G6PD” to “G6PDH” in line 611

Our responses: Thank you very much for pointing out. Although both “G6PD” and “G6PDH” are used for abbreviated designation of glucose-6-phosphate dehydrogenase and mentioned so in the title of reference 175, we have changed “G6PD” to “G6PDH”, according to the advice.

 Correct “ferropsosis” in the line 693

Our responses: Thank you very much for pointing out. We have corrected it.

Reviewer 2 Report

Comments and Suggestions for Authors

Review on the manuscript of Fujii J & Imai H: “Oxidative metabolism as a cause of lipid peroxidation in the execution of ferroptosis”.

In this study, the Authors review the literature on the role of oxidative metabolism in ferroptosis.

Overall, I found this topic to be of great interest, as ferroptosis represents a recently elucidated type of cell death with significant implications for various diseases. Therefore, it is essential to understand the role of fundamental biological processes, such as oxidative metabolic pathways, in ferroptosis induction. This understanding may facilitate the development of potential therapeutic targets to modulate ferroptosis under specific pathological conditions.

In my opinion, the manuscript is well-structured, and the Authors have effectively addressed the main question proposed. However, there are some issues that I would like to highlight in the current form of the manuscript. I hope the Authors find the following comments and suggestions useful.

1 - On line 38, the Authors mention that "Ferroptosis is an iron-dependent necrotic cell death...". I recommend modifying the sentence to "Ferroptosis is an iron-dependent nonapoptotic cell death", as ferroptosis is not a type of necrotic cell death, but rather a distinct form of cell death independent of necrosis.

2 - Figure 1 depicts several mechanistic pathways that may lead to ferroptosis. To make the figure more understandable, I recommend that the Authors include a sequence of numbers in the figure and describe these steps in the legend. This way, it will be easier for readers to comprehend the figure. The same idea would be beneficial for figures 3 and 4. In addition, I recommend that the Authors include in the figure legend descriptions of all abbreviations used. It will help the readers.

3 - In section 4.4. Iron-independent lipid peroxidation induced cell death (Lipoxytosis) regulated by GPX4, the Authors indicate that “Although MEK1/2 inhibitors suppressed both GPX4-deficient cell death and RSL3- or erastin-induced ferroptosis, GPX4-deficient cell death was MEK1-dependent but RSL3- or erastin-induced ferroptosis was not”. It is not clear why MEK1/2 inhibitors suppress RSL3- or erastin-induced ferroptosis and this effect is independent of MEK1. I would recommend that the Authors clarify this point.

4 - Since the manuscript focuses on oxidative metabolism as a cause of lipid peroxidation in the execution of ferroptosis, it would be beneficial to include a section on the key metabolic pathways involved in oxidative metabolism (e.g., Krebs cycle, electron transport chain), highlighting how ROS are generated in these pathways.

5 - I also recommend that the Authors include a section on future perspectives in the manuscript. In this section, the Authors could provide ideas for future exploration, emerging technologies, and approaches to study the involvement of oxidative metabolism in ferroptosis, among other aspects. In my opinion, review manuscripts benefit from this type of discussion.

6 - Throughout the manuscript, some English issues were detected. If the Authors have the possibility to have the manuscript proofread by a native English speaker, it would be beneficial.

Comments on the Quality of English Language

Minor editing of English language required.

Author Response

Thank you very much for your efforts on reviewing and criticism on our manuscript. We are grateful to the reviewers for their valuable comments that helped improve the manuscript. Our individual responses follow after their comments.

Reviewer 2

Suggestions for Authors

Review on the manuscript of Fujii J & Imai H: “Oxidative metabolism as a cause of lipid peroxidation in the execution of ferroptosis”.

In this study, the Authors review the literature on the role of oxidative metabolism in ferroptosis.

Overall, I found this topic to be of great interest, as ferroptosis represents a recently elucidated type of cell death with significant implications for various diseases. Therefore, it is essential to understand the role of fundamental biological processes, such as oxidative metabolic pathways, in ferroptosis induction. This understanding may facilitate the development of potential therapeutic targets to modulate ferroptosis under specific pathological conditions.

In my opinion, the manuscript is well-structured, and the Authors have effectively addressed the main question proposed. However, there are some issues that I would like to highlight in the current form of the manuscript. I hope the Authors find the following comments and suggestions useful.

1 - On line 38, the Authors mention that "Ferroptosis is an iron-dependent necrotic cell death...". I recommend modifying the sentence to "Ferroptosis is an iron-dependent nonapoptotic cell death", as ferroptosis is not a type of necrotic cell death, but rather a distinct form of cell death independent of necrosis.

Our responses: Thank you very much for valuable comment. According to your kind advice, we have changed the word in Introduction and Abstract. 

2 - Figure 1 depicts several mechanistic pathways that may lead to ferroptosis. To make the figure more understandable, I recommend that the Authors include a sequence of numbers in the figure and describe these steps in the legend. This way, it will be easier for readers to comprehend the figure. The same idea would be beneficial for figures 3 and 4. In addition, I recommend that the Authors include in the figure legend descriptions of all abbreviations used. It will help the readers.

Our responses:  Thank you for the comment on figures and their legends. Concerning number on the reaction sequence, metabolic reactions do not flow in one direction but are interrelated, and many reactions are reversible, so it is impossible to number in one direction them in these figures following the reviewer’s request. What the reviewers requested, i.e. direction of metabolic flows, is explained in the text, so including them in the legend would be duplicated description. Schematic diagrams are intended to aid in comprehension of the text and are not intended that the entire content be understood from the figures alone. Therefore, we decided not to adopt the reviewers' suggestion on revising the figures and legends.

Concerning abbreviations, technical terms are generally given in full with its abbreviation in parentheses when first mentioned, and thereafter only the abbreviations are given. Readers reads text first and then look at figures, because they are provided to help understand the descriptions in the text, but not the other way around. Therefore, we wrote full spelling of abbreviations to the legend only when they are first appeared, being the text primary, as most journals do.

3 - In section 4.4. Iron-independent lipid peroxidation induced cell death (Lipoxytosis) regulated by GPX4, the Authors indicate that “Although MEK1/2 inhibitors suppressed both GPX4-deficient cell death and RSL3- or erastin-induced ferroptosis, GPX4-deficient cell death was MEK1-dependent but RSL3- or erastin-induced ferroptosis was not”. It is not clear why MEK1/2 inhibitors suppress RSL3- or erastin-induced ferroptosis and this effect is independent of MEK1. I would recommend that the Authors clarify this point.

Our responses: Thank you for your comment. It is known that MEK normally phosphorylates ERK. Ferroptosis induced by RSL3 and erastin was inhibited by MEK1/2 inhibitors, but not by ERK1/2 inhibitors or ERK1 and ERK2 selective inhibitors. This suggests that normal MEK-ERK signaling is not involved in ferroptosis, and although we cannot exclude the possibility that the inhibitory effect of MEK1/2 inhibitors on ferroptosis is mediated solely by MEK2, it is possible that inhibition of MAPKK other than MEK may inhibit ferroptosis.We have changed the text as follows.

4 - Since the manuscript focuses on oxidative metabolism as a cause of lipid peroxidation in the execution of ferroptosis, it would be beneficial to include a section on the key metabolic pathways involved in oxidative metabolism (e.g., Krebs cycle, electron transport chain), highlighting how ROS are generated in these pathways.

Our responses:  Thank you for valuable comment. We understand your point, but there are many excellent reviews on production of ROS from these well-established metabolic pathways. Therefore, the TCA cycle and electron transport chain are not overviewed again in our article, but the process in which radical species are generated in these pathways was described at corresponding point, indeed. We would like to avoid going into details of the established processes any further but rather focus on the issue how elevated oxidative metabolisms associate with ferroptosis.

5 - I also recommend that the Authors include a section on future perspectives in the manuscript. In this section, the Authors could provide ideas for future exploration, emerging technologies, and approaches to study the involvement of oxidative metabolism in ferroptosis, among other aspects. In my opinion, review manuscripts benefit from this type of discussion.

Our responses: We agree about writing a future perspective. It is simple statement, but the following sentence at the end of Concluding remark may correspond the subject. “Identification of the source of the causative radical species could contribute development of effective therapeutics for ferroptosis-related diseases.” This review article discusses oxidative metabolism as a potential cause of ferroptosis, which is a perspective that has received little attention before. Therefore, we believe that this article itself proposed new perspectives for the future of ferroptosis research and clinical application. Accordingly, although there is no session on future perspective, we did provide a future outlook by this article.

6 - Throughout the manuscript, some English issues were detected. If the Authors have the possibility to have the manuscript proofread by a native English speaker, it would be beneficial.

Our responses:  The manuscript was edited by a native English speaker before submission, as stated in the attached certificate. We checked the manuscript again before re-submission. We hope the English issue has been resolved now.